# Mechanism of chiral proofreading during translation of the genetic code

Sadeem Ahmad[†], Satya Brata Routh[†], Venu Kamarthapu, Jisha Chalissery, Sowndarya Muthukumar, Tanweer Hussain, Shobha P Kruparani, Mandar V Deshmukh, Rajan Sankaranarayanan*

Structural Biology Laboratory, Centre for Cellular and Molecular Biology, Council for Scientific and Industrial Research, Hyderabad, India

**Abstract** The biological macromolecular world is homochiral and effective enforcement and perpetuation of this homochirality is essential for cell survival. In this study, we present the mechanistic basis of a configuration-specific enzyme that selectively removes D-amino acids erroneously coupled to tRNAs. The crystal structure of dimeric D-aminoacyl-tRNA deacylase (DTD) from *Plasmodium falciparum* in complex with a substrate-mimicking analog shows how it uses an invariant 'cross-subunit' Gly-*cis*Pro dipeptide to capture the chiral centre of incoming D-aminoacyl-tRNA. While no protein residues are directly involved in catalysis, the unique side chain-independent mode of substrate recognition provides a clear explanation for DTD's ability to act on multiple D-amino acids. The strict chiral specificity elegantly explains how the enriched cellular pool of L-aminoacyl-tRNAs escapes this proofreading step. The study thus provides insights into a fundamental enantioselection process and elucidates a chiral enforcement mechanism with a crucial role in preventing D-amino acid infiltration during the evolution of translational apparatus.

*For correspondence: sankar@ccmb.res.in

[†]These authors contributed equally to this work

Competing interests: The authors declare that no competing interests exist.

## Introduction

The origin of homochirality in biological macromolecules has been a subject of active research and intense debate till date (*Podlech, 2001*; *Blackmond, 2010*). With the selection of only L-amino acids (L-aas) for incorporation in proteins, effective enforcement and perpetuation of homochirality became essential for an efficient translational machinery to be a part of living systems. To this end, multiple checkpoints ensure that only L-aas are incorporated during translation. These include aminoacyl-tRNA synthetases (aaRSs), elongation factor Tu (EF-Tu) and ribosome (*Jonak et al., 1980*; *Pingoud and Urbanke, 1980*; *Bhuta et al., 1981*; *Yamane et al., 1981*; *Ban et al., 2000*; *Agmon et al., 2004*; *Ogle and Ramakrishnan, 2005*). Many aaRSs possess proofreading modules that remove similar non-cognate L-aas mistakenly attached to tRNAs and thus ensure fidelity of translation (*Nureki et al., 1998*; *Silvian et al., 1999*; *Dock-Bregeon et al., 2004*). However, a freestanding enzyme D-aminoacyl-tRNA deacylase (DTD) removes D-amino acids (D-aas) mischarged on tRNAs and ensures that D-aas do not get incorporated into proteins (*Calendar and Berg, 1967*; *Wydau et al., 2009*; *Zheng et al., 2009*). Since DTDs act in *trans* as freestanding modules, they are most likely to operate through resampling by recapturing aminoacyl-tRNAs (aa-tRNAs) from EF-Tu (*Ling et al., 2009*).

A DTD-like fold has been found appended to archaeal threonyl-tRNA synthetase (ThrRS) where it removes mischarged L-serine from tRNA[Thr] (*Dwivedi et al., 2005*; *Hussain et al., 2006*, *2010*). The structure of archaeal ThrRS editing domain from *Pyrococcus abyssi* (Pab-NTD) not only highlighted the evolutionary link between DTD and Pab-NTD but also suggested the probable role this fold might have played in enforcement of homochirality during early evolution of translational machinery, since weakly discriminating primordial aaRSs would have been less enantioselective (*Dwivedi et al., 2005*). Even some of the highly evolved present day aaRSs have been shown to be inherently weak in enantioselection,

**eLife digest** Amino acids are 'chiral' molecules that come in two different forms, called D and L, which are mirror images of each other, similar to how our left and right hands are mirror images of each other. However, only one of these forms is used to make proteins: the more abundant L-amino acids are linked together to make proteins, whereas the scarcer D-amino acids are not. This 'homochirality' is common to all life on Earth.

The molecular machinery inside cells that manufactures proteins involves many enzymes that carry out different tasks. Among these is an enzyme called DTD (short for D-aminoacyl-tRNA deacylase), which prevents D-amino acids being incorporated into proteins. To do this, DTD must be able to recognise and remove the D forms of many different amino acids before they are taken to the growing protein by transfer RNA molecules. However, the details of this process are not fully understood.

To investigate this mechanism, Ahmad et al. made crystals of the DTD enzyme in complex with a molecule that mimics a D-amino acid attached to a transfer RNA molecule. By studying this structure at a high resolution, Ahmad et al. were able to identify how the active site of DTD can specifically accommodate the 'chiral centre' of a complex made of a D-amino acid and a transfer RNA molecule.

DTD is able to recognize D-amino acids because of a critical dipeptide that is inserted from one subunit of the DTD into the active site of another subunit of the enzyme. The effect of this dipeptide is to generate a binding pocket that is a perfect fit for the chiral centre of a complex that contains a D-amino acid and a transfer RNA molecule. Moreover, this pocket specifically excludes complexes that contain an L-amino acid.

The crucial parts of DTD that form the binding pocket are highly conserved—that is, they are the same in a wide variety of organisms, from bacteria to mammals. This conservation suggests that DTD is crucial for ensuring homochirality throughout all forms of life. Intriguingly, DTD is particularly highly expressed in neurons which are abundant in D-amino acids: this indicates that the DTD enzyme has an important physiological role, which will certainly be the focus of future work.

leading to the formation of D-aminoacyl-tRNAs (D-aa-tRNAs) (*Calendar and Berg, 1966*; *Soutourina et al., 2000b*). D-aa-tRNAs thus formed could either get incorporated into the growing polypeptide chain leading to global misfolding or get accumulated in the cell leading to depletion of tRNA pool. Either way, decoupling of D-aa from tRNA is extremely important which makes the cellular role of DTD crucial.

DTD activity was originally identified in 1967 by Calendar and Berg and the function is conserved in all organisms including humans (*Calendar and Berg, 1967*; *Zheng et al., 2009*). So far three distinct types of DTDs have been reported. The most commonly found canonical DTD has been shown to be present in most bacteria and all eukaryotes (*Soutourina et al., 1999*). Archaea, on the other hand, lack canonical DTD sequence in their genomes and instead possess another structurally unrelated protein which carries out the function of deacylating D-aa-tRNAs (*Ferri-Fioni et al., 2006*). This functional equivalent of DTD has been termed DTD2 and it is found in archaea and plants (*Wydau et al., 2007*). The third type of DTD, known as DTD3, has been reported in some cyanobacteria that lack both canonical DTD and DTD2 (*Wydau et al., 2009*). Overall, the universal distribution of DTD function across the three domains of life clearly suggests an essential role DTDs must have played and continue to play in enforcing homochirality. From here on, DTD would refer to the canonical DTD found in bacteria and eukaryotes unless otherwise mentioned. The DTD sequence is highly conserved among prokaryotes and eukaryotes with the sequence identity between *Escherichia coli* and *Homo sapiens* being 39%. The biological significance of DTD has been shown in both prokaryotes and eukaryotes with deletion of *dtd* gene leading to reduced tolerance to several D-aas in a dose-dependent manner (*Soutourina et al., 2000a*, *2000b*, *2004*; *Zheng et al., 2009*). DTD is ubiquitously expressed and shows high levels of expression in the human neuronal cells, which are abundant in D-aas, thus strongly indicating a critical role of DTD (*Zheng et al., 2009*).

Mechanistically, the most remarkable challenge that DTD faces is to specifically act on multiple D-aa-tRNAs while rejecting L-aminoacyl-tRNAs (L-aa-tRNAs) without any specificity for either the amino acid or the tRNA. This can be seen from the fact that DTD is able to act on diverse substrates such as Tyr, Phe, Asp, and Trp as long as they carry a D-configuration of the amino acid on tRNA (*Calendar*

*and Berg, 1967*; *Soutourina et al., 2000b*). The problem is further compounded by the very high excess of L-aa-tRNA over D-aa-tRNA in the cellular milieu and warrants a stringent D-configuration specificity to avoid depletion of L-aa-tRNA pool. Although biochemical studies have indicated its configurational preference, the mechanistic basis of this fundamental process remained elusive due to lack of a cognate substrate-bound complex structure.

The first crystal structure of DTD from *E. coli* (*Ec*DTD) was solved in the apo form, which identified this novel DTD-like fold (*Ferri-Fioni et al., 2001*). Later, the apo structures of DTD from *Haemophilus influenzae* (*Lim et al., 2003*), *Aquifex aeolicus* (PDB id: 2DBO) and *H. sapiens* (*Kemp et al., 2007*) also became available. In the absence of any ligand-bound structure, docking studies were done with *H. influenzae* DTD in an attempt to understand its mechanism (*Lim et al., 2003*). Recently, the structure of *Plasmodium falciparum* DTD (*Pf*DTD) was solved in complex with ADP and multiple free D-aas (*Bhatt et al., 2010*). Although these studies had proposed a catalytic mechanism implicating the role of a Thr residue, the structural basis of DTD's strict enantioselectivity was not clear. In this study, we report the mechanism of this crucial process with the help of high resolution structures of *Pf*DTD in complex with a substrate-mimicking analog. We further validate the mechanistic proposal with the help of biochemical assays conducted on *Pf*DTD as well as *Ec*DTD and NMR-based binding studies with *Pf*DTD. The work identifies the essential role of a universally conserved 'cross-subunit' Gly-*cis*Pro motif in providing exclusive enantioselectivity to the enzyme thus ensuring homochirality during translation.

## Results

### Co-crystal structure of *Pf*DTD with D-Tyr3AA

*Pf*DTD was co-crystallized with a post-transfer substrate analog D-Tyr3AA, which mimics D-tyrosine attached to the 3'-OH of the terminal adenosine (A76) of tRNA (*Figure 1*). The ester linkage between amino acid and adenosine is replaced by an amide linkage to make it non-hydrolyzable. Similar post-transfer substrate analogs have been used extensively to study proofreading mechanisms in atomic details for both Class I-specific CP1 editing domains and Class II-specific editing domains (*Lincecum*

**D-Tyr-tRNA^Tyr**
**(Actual substrate)**

**D-Tyr3AA**
**(Substrate analog)**

**Figure 1**. Comparison of the actual substrate with the analog used in this study. (★) The 5'-OH is linked to tRNA in the actual substrate, whereas it is free in D-Tyr3AA. (★) The ester bond in the real substrate is replaced by an amide bond in the analog D-Tyr3AA to make it non-hydrolyzable.

The following figure supplements are available for figure 1:

**Figure supplement 1**. Stereoscopic images showing the positions of earlier modeled ligands with respect to the cognate substrate-mimicking analog D-Tyr3AA in DTD.

**Figure supplement 2**. The reported structures of proofreading domains with substrate-mimicking analogs.

**Table 1.** Crystallographic data collection and refinement statistics

| | PfDTD+D-Tyr3AA | PfDTD+D-Tyr3AA |
| --- | --- | --- |
| | Crystal I | Crystal II |
| | (PDB id: 4NBI) | (PDB id: 4NBJ) |
| Data Collection | | |
| Space group | C2 | P2$_1$ |
| Cell dimensions: | | |
| a (Å) | 82.13 | 90.90 |
| b (Å) | 65.74 | 79.91 |
| c (Å) | 56.92 | 95.02 |
| β (°) | 93.30 | 93.51 |
| Resolution range (Å)* | 25.0–1.86 (1.93–1.86) | 25.0–2.20 (2.28–2.20) |
| Total Observations | 178996 | 449245 |
| Unique reflections | 25156 (2283) | 69275 (6911) |
| Completeness (%) | 98.3 (89.3) | 100 (99.9) |
| R$_{merge}$ (%) | 7.5 (28.3) | 11.8 (59.3) |
| $\langle I/(\sigma)I \rangle$ | 29.4 (5.5) | 17.7 (2.8) |
| Redundancy | 7.1 (6.4) | 6.5 (5.9) |
| Data refinement | | |
| Resolution (Å) | 1.86 | 2.20 |
| No. of reflections | 23878 | 65743 |
| R (%) | 16.77 | 19.46 |
| R$_{free}$ (%)† | 19.25 | 25.35 |
| Monomers/a.u. | 2 | 8 |
| No. of residues | 323 | 1289 |
| No. of atoms | 2917 | 10727 |
| Protein | 2595 | 10104 |
| Ligand | 72 | 248 |
| Water | 250 | 375 |
| R.m.s. deviation | | |
| Bond lengths (Å) | 0.007 | 0.010 |
| Bond angles (°) | 1.093 | 1.436 |
| Mean B value (Å$^2$) | 30.33 | 48.11 |
| Protein | 29.09 | 47.69 |
| Ligand | 43.94 | 53.30 |
| Water | 41.29 | 57.16 |

*Values in parentheses are for the highest resolution shell.
†Throughout the refinement, 5% of the total reflections were held aside for R$_{free}$.

*et al., 2003*; *Dock-Bregeon et al., 2004*; *Fukunaga and Yokoyama, 2006*; *Hussain et al., 2006*, *2010*). The crystal structure of *Pf*DTD in complex with D-Tyr3AA has been solved in two different crystal forms: crystal form I at a resolution of 1.86 Å in C2 space group and crystal form II at a resolution of 2.2 Å in P2$_1$ space group (*Table 1*). Crystal forms I and II have two and eight copies per asymmetric unit, respectively. This provides us with 10 independent observations of the ligand in the active site (*Figure 2—figure supplement 1*). Since all copies present a similar picture, the higher resolution crystal form I is discussed here unless otherwise mentioned (*Figure 2A*). The enzyme is a symmetric dimer with two active sites per dimer that are located at the dimeric interface (*Figure 2B,C*). The residues defining the active site pocket span the conserved –SQFTL– motif from one monomer and the –NXGP(V/F)T– motif from the other. The D-Tyr3AA-bound structure superimposes on the apo structure (PDB id: 3KNF) with an r.m.s.d. of 0.41 Å for 260 Cα atoms (*Figure 2—figure supplement 2*). However, there are subtle rearrangements of the active site region upon ligand binding, indicating the plasticity associated with the active site (*Figure 2D*). The most noticeable movements occur in Phe89, Phe137 and Gly138 upon accommodation of D-Tyr3AA making the active site more compatible for substrate binding (*Figure 2D*).

## Adenosine binding and catalytic mechanism

The active site of DTD uses, in a major way, the main chain atoms to interact with the substrate (*Figure 2E*). The main chain atoms of Lys107 and Ile43 have direct and water-mediated interactions with the adenine moiety. An invariant Phe137 provides base-stacking interaction to the adenine ring. The main chain nitrogen of Gly138 along with the side chain hydroxyl of Ser87 holds the 2'-OH. The 5'-OH projects outwards as should be expected since it would be attached to the preceding nucleotide (C75) in the actual substrate, which is D-aa-tRNA. Considering that Pab-NTD, which is a structural homolog of DTD (*Figure 2—figure supplement 3*), also interacts with the substrate mostly through main chain atoms, it appears to be a conserved feature of this fold to employ main chain atoms extensively for ligand binding (*Figure 2—figure supplement 4*) (*Hussain et al., 2006*, *2010*). Moreover, the adenosine-binding pocket is highly conserved in this DTD-like fold with an invariant Phe providing base-stacking interaction (Phe117 in Pab-NTD and Phe137 in *Pf*DTD) as shown in *Figure 2—figure supplement 4*. To prove that the ligand complex we have obtained is a biologically relevant one, we disrupted the adenine-binding pocket with the help of mutations and showed that it leads to complete loss of activity. As shown in *Figure 3A*, Phe137 that stacks with the

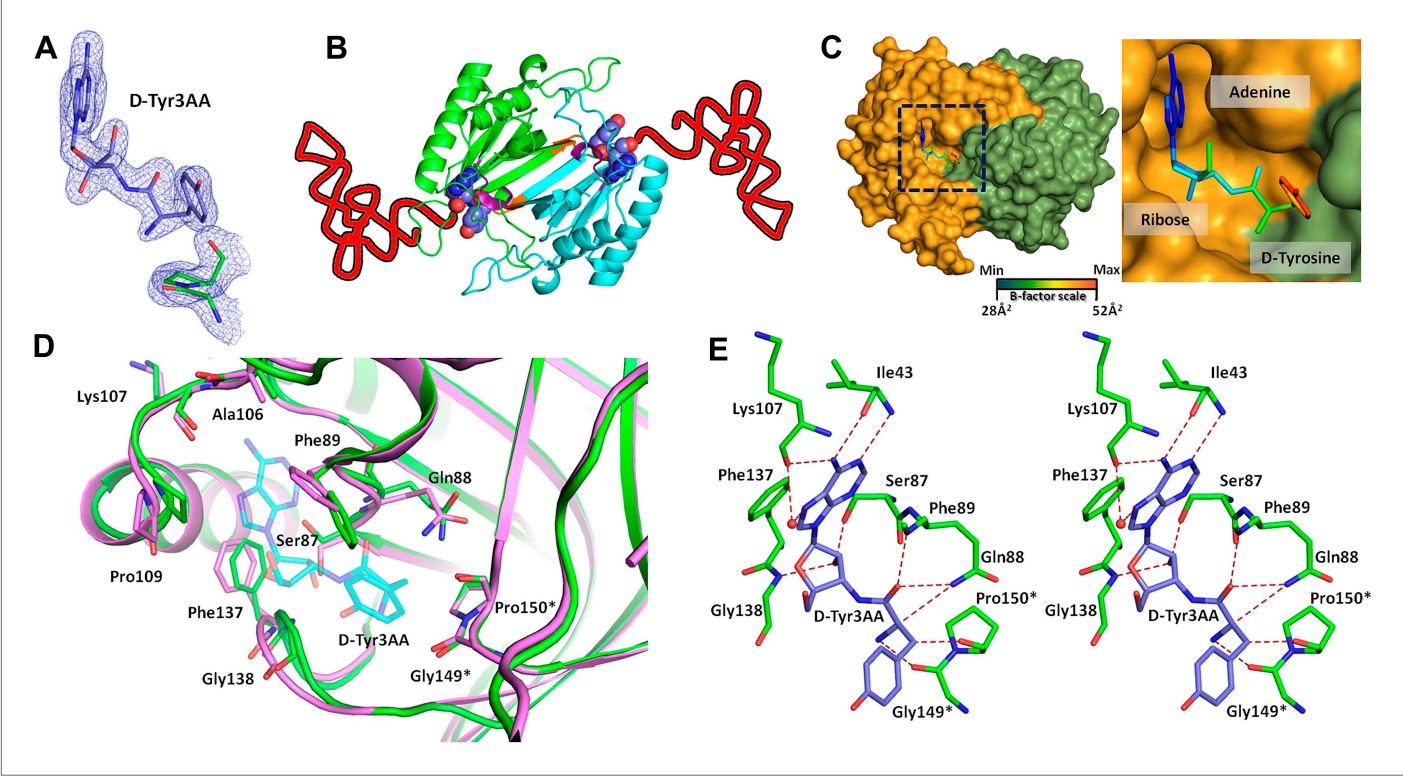

**Figure 2**. Structure of DTD in complex with D-Tyr3AA. (**A**) A (2Fo–Fc) map contoured at 1.2σ clearly showing unambiguous density for the ligand D-Tyr3AA from crystal form I solved at 1.86 Å resolution. (**B**) Dimeric DTD with the two monomers shown in green and cyan. The conserved–SQFTL–and–NXGP(V/F)T–motifs are depicted in violet and orange respectively. The ligand binds in the two active sites located at the dimer interface. The two tRNAs have been schematically represented. (**C**) Surface representation showing D-Tyr3AA in the pocket. Inset is a magnified image showing the side chain of D-tyrosine protruding out of the pocket. The ligand has been colored according to the B-factors. (**D**) Structural rearrangements in the substrate pocket upon D-Tyr3AA binding highlighting the plasticity of the active site. The apo is shown in green and the complex is shown in purple. The ligand has been made transparent for clarity. (**E**) Stereoscopic representation showing the interactions between the ligand and the active site residues (* indicates residues from the other monomer).

The following figure supplements are available for figure 2:

**Figure supplement 1**. Electron density for the ligand in all observations.

**Figure supplement 2**. Superimposition of D-Tyr3AA-bound complex structure of *Pf*DTD (pink) on the apo structure (green).

**Figure supplement 3**. Superimposition of *Pf*DTD on Pab-NTD.

**Figure supplement 4**. Comparison of ligand interaction in *Pf*DTD and Pab-NTD.

**Figure supplement 5**. Atomic B-factors plotted for all the ligand atoms from both crystal forms I and II.

**Figure supplement 6**. Superimposition of D-Tyr3AA from all monomers of crystal forms I and II.

adenine base was mutated to Ala. In another mutant, we blocked the adenine pocket by mutating a conserved Ala112 to a bulkier Phe (***Figure 3A***). Both F137A and A112F mutations resulted in a complete loss of activity, confirming that the adenosine-binding pocket identified here indeed represents the bona fide functional site (***Figure 3B***). The corresponding mutations F125A and A102F in *Ec*DTD were also tested for their activity against D-Tyr-tRNA^Tyr. These mutants in *Ec*DTD also showed a complete loss of activity (***Figure 3—figure supplement 1B***), further substantiating the biological relevance of the substrate-binding pocket identified here.

To delineate the catalytic mechanism, we looked for all the amino acid side chains located within a distance of 6 Å from the susceptible bond of the substrate, that is the bond between adenosine and

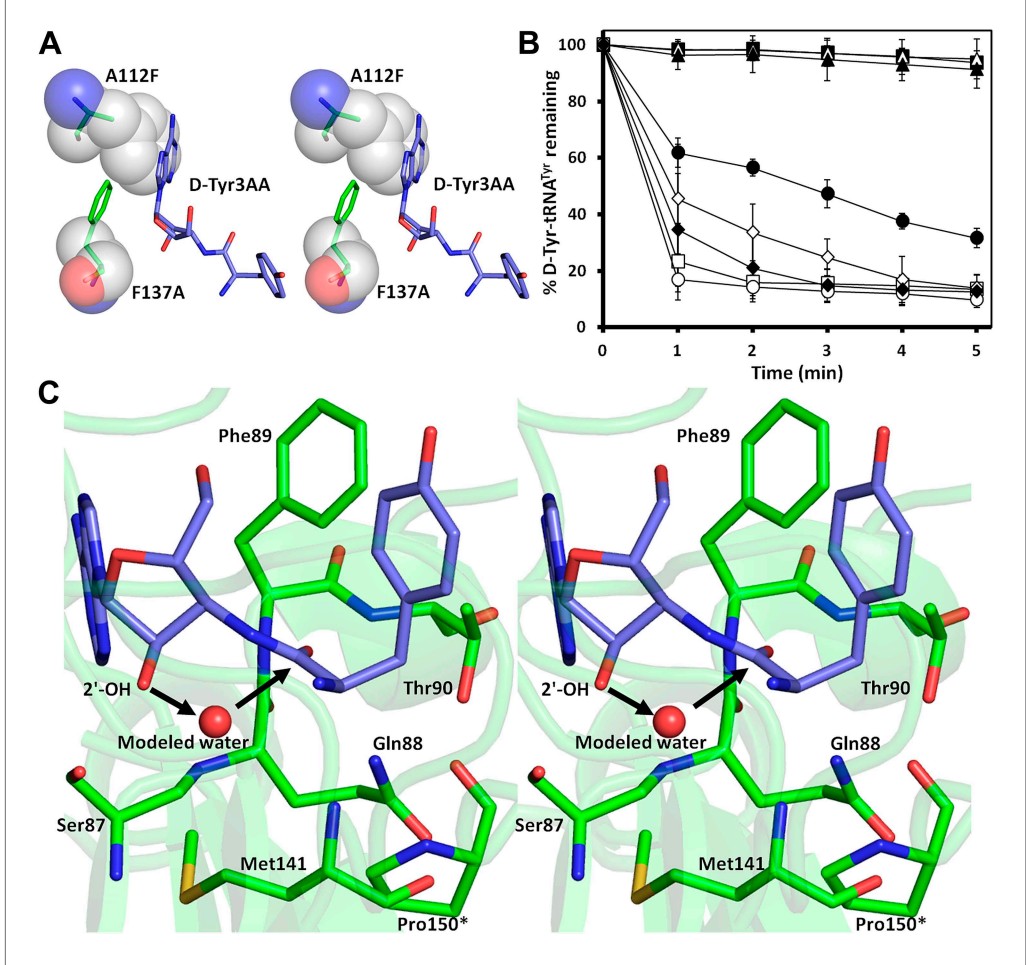

**Figure 3**. Mutational analysis of the active site residues. (**A**) Stereoscopic depiction showing mutations generated in the adenine-binding pocket: Stick representation is used for wild-type residues while mutants are depicted in spheres. Phe137 was mutated to Ala and Ala112 was mutated to Phe. (**B**) Deacylation of D-Tyr-tRNA$^{Tyr}$ by buffer (—■—), wild-type *Pf*DTD (—□—), F137A (—▲—), A112F (—△—), S87A (—●—), S87P (—○—), Q88A (—◇—) and T90A (—◆—). 500 pM enzyme concentration was used for the assays. (**C**) Stereoscopic image showing all the protein side chains within 6 Å of the susceptible bond of the substrate. A water molecule has been modeled based on Pab-NTD complex structure. The water is positioned at a distance of 2.61 Å from the 2'-OH and 2.79 Å from the scissile bond of D-Tyr3AA. In the absence of any protein side chain playing a role in catalysis, a substrate-assisted mechanism is proposed involving the role of 2'-OH of tRNA in activating a water molecule as suggested in case of Pab-NTD.

The following figure supplements are available for figure 3:

**Figure supplement 1**. Mutational analysis of the active site residues in *Pf*DTD and *Ec*DTD.

**Figure supplement 2**. Mutational data on the earlier identified binding modes in DTD.

the carbonyl group of D-tyrosine. These residues include Ser87, Gln88, Phe89, Thr90, Met141, and Pro150. Out of these, the residues that can chemically contribute to catalysis are Ser87, Gln88, and Thr90, which are positioned at a distance of 5.71 Å, 3.56 Å, and 5.72 Å respectively from the carbonyl carbon of the substrate (***Figure 3C***).

To probe the role played by these residues in catalysis, we generated mutants S87A, S87P, Q88A, Q88N, Q88E, T90A, and T90S, and tested them for deacylation activity. All mutants deacylated D-Tyr-tRNA$^{Tyr}$ as efficiently as the wild type *Pf*DTD, except S87A that showed partly compromised activity (***Figure 3B***, ***Figure 3—figure supplement 1A***). Although S87A was only moderately active, the fact that S87P retains complete activity rules out any catalytic role for this residue. Therefore, even though

Ser87 interacts with 2'-OH of the ribose, it seems to perform a space-filling function of maintaining the ribose in an active conformation. It is also worth noting here that in some DTDs from different organisms, Ser87 is naturally substituted by a Pro, which further proves that the side chain chemistry of this residue is not essential for catalysis. The catalytic role of other protein residues Gln88 and Thr90 can also be ruled out as Q88A, Q88N, Q88E, T90A, and T90S mutants deacylated D-Tyr-tRNA$^{Tyr}$ as efficiently as the wild type (*Figure 3B*, *Figure 3—figure supplement 1A*). Strikingly, Thr90 was identified from the modeling studies (*Lim et al., 2003*) as a crucial residue responsible for catalysis, as discussed further in a later section. However, mutating this residue did not at all affect the activity of the enzyme. The corresponding mutants S77A, S77P, Q78A, and T80A in *Ec*DTD were also tested for their deacylation activity. In the case of *Ec*DTD, all mutants including S77A deacylated D-Tyr-tRNA$^{Tyr}$ as efficiently as the wild type (*Figure 3—figure supplement 1B*). The above data suggest that none of the protein residues around the scissile bond are involved in catalysis.

Our earlier structural studies on Pab-NTD have suggested an RNA-assisted catalytic mechanism implicating the role of 2'-OH in activating a water molecule for catalysis (*Hussain et al., 2006*, *2010*). Subsequently, the catalytic role of RNA in proofreading has also been experimentally shown in the case of phenylalanyl-tRNA synthetase (PheRS) (*Ling et al., 2007*). Unlike in the case of PheRS, the catalytic role of RNA in DTD could not be directly probed with a modified tRNA having a terminal 2'-deoxyadenosine since tyrosyl-tRNA synthetase (TyrRS) attaches the amino acid on 2'-OH of the ribose, which is then transesterified to 3'-OH for proofreading reaction. As we show later, this transesterification is required for DTD to act since it is expected to recognize aminoacyl moiety only when it is attached to the 3'-OH. A comparison of non-cognate and cognate substrate analog-bound structures of Pab-NTD had revealed that the space available in the reaction zone is crucial for catalysis. It was shown that upon cognate substrate binding this space is constricted due to a subtle movement of a crucial Lys side chain (*Hussain et al., 2010*). This limited space, therefore, does not allow the putative catalytic water molecule to be accommodated in that site as it would have serious short contacts, and hence no deacylation. Although we do not observe a water molecule in that region in DTD, there is enough space available for a water molecule to be positioned without any clashes. Furthermore, it is worth noting here that the site of catalysis in DTD is much more accessible to the external bulk solvent as compared to Pab-NTD and could be a plausible reason as to why we do not observe the water molecule crystallographically. Therefore, considering the structural similarity and conservation of substrate-binding modes between DTD and Pab-NTD along with the experimental evidence showing the absence of any direct role of protein side chains in the catalytic mechanism, we propose a similar RNA-assisted catalysis in DTD also (*Figure 3C*). The 2'-OH of the terminal ribose would activate a water molecule, which in turn makes nucleophilic attack on the carbonyl carbon of the substrate. The resultant tetrahedral transition state would be stabilized by the oxyanion hole formed by main chain nitrogen atoms of Phe89 and Thr90 situated at a distance 3.03 Å and 4.05 Å respectively from the carbonyl oxygen of the substrate. It would then result in the subsequent cleavage of the ester bond between the D-aa and the tRNA. Therefore, taken together with studies on Pab-NTD and the primordial nature of its fold and function, the above data indicate that the DTD fold is an RNA-based catalyst in the proofreading reaction.

## Enantioselection mechanism

A striking feature of the amino acid recognition site is the capture of all the atoms attached to the chiral centre Cα and the role of cross-subunit interactions, particularly a Gly-*cis*Pro motif from both monomers inserted into the active site of the dimeric counterpart that plays a central role in the recognition mechanism, as described in 'Mechanism of L-amino acid rejection from the active site'. The aminoacyl moiety has interactions with residues from both monomers. The carbonyl oxygen interacts with the main chain nitrogen of Phe89 and the side chain amide of Gln88. Both the residues belong to the –SQFTL– motif. The α-amino group of D-tyrosine has an interaction with carbonyl oxygen of Gly149 from the cross-subunit Gly-*cis*Pro motif. Such a capture of the carbonyl oxygen and the amino group of the incoming D-aa, automatically positions the Cβ in such a way that it makes favorable C-H⋯O hydrogen bond with the carbonyl oxygen of Pro150, again from the cross-subunit Gly-*cis*Pro motif. In addition, the Cα also makes a weak C-H⋯N bond with the Gln88 side chain amide nitrogen. The interaction distances of the aminoacyl moiety have been summarized in *Supplementary file 1A*. With this mode of recognition of the configuration, the side chain of D-tyrosine is positioned in such a way that it projects out of the binding pocket and has no interaction beyond the Cβ atom as seen in

*Figure 2C*. The atomic B-factors of the ligand clearly show a sharp rise in the side chain atoms beyond the Cβ (*Figure 2C*, *Figure 2—figure supplement 5*, *Supplementary file 1B*). The superimposition of all the copies of ligand from both the crystal forms I and II shows considerable deviations in only the side chain atoms beyond Cβ (*Figure 2—figure supplement 6*). The lack of recognition of side chain atoms indicates that residues with different side chain chemistries and sizes are treated alike. Such a side chain-free recognition mechanism provides the basis for how nature has designed a single deacylase to deal with any D-aa-tRNA and reveals the crucial role played by weak hydrogen bonds in D-chirality selection.

## Mechanism of L-amino acid rejection from the active site

If an L-aa was to bind in this pocket, it would have to do so in one of the three theoretically possible conformations shown in *Figure 4*. In conformation I, where the side chain swaps positions with Hα, it would result in serious clashes with several atoms in the binding pocket (*Figure 4C*). Even the Cβ of L-Tyr would have short contacts of 3.08 Å with the Cδ and 2.69 Å with the carbonyl oxygen of Pro150. In conformation II, the side chain would occupy the place of the amino group (*Figure 4D*). In this position it would be placed adjacent to 5'-OH and would therefore have short contacts with the preceding

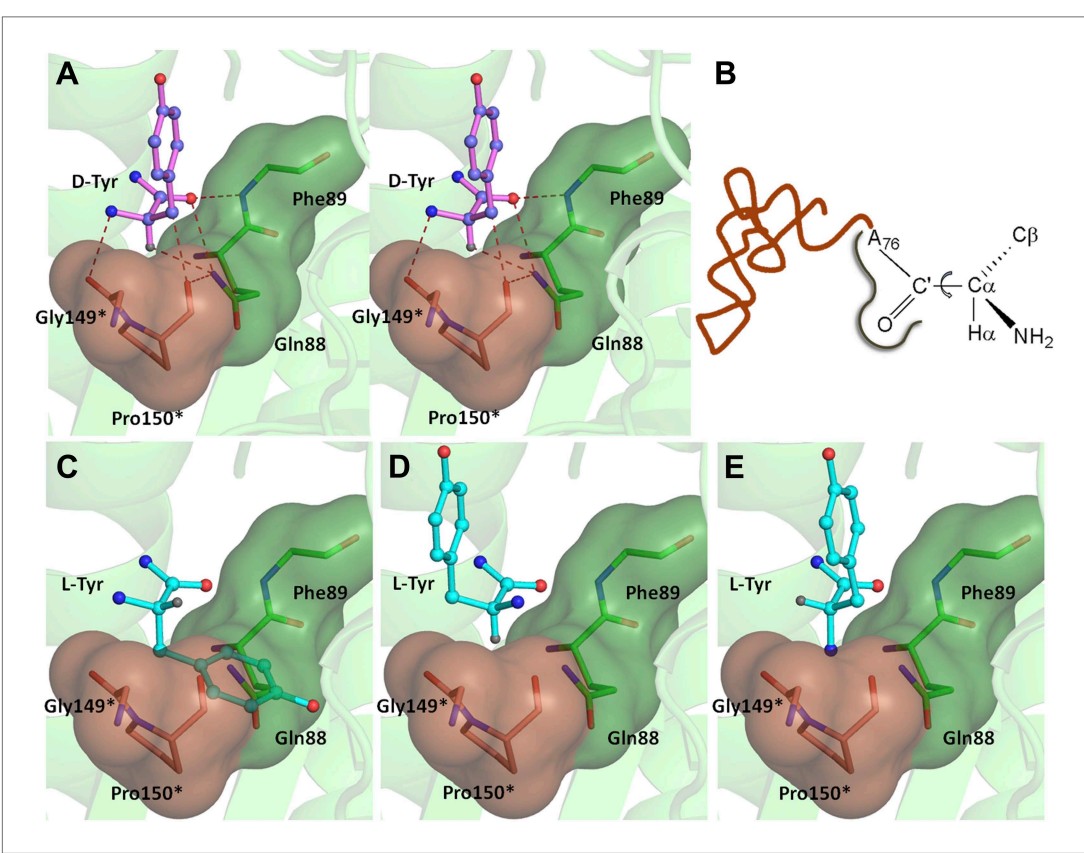

**Figure 4**. Mechanism of L-chirality rejection. The cross-subunit Gly-*cis*Pro motif is shown in brown. (**A**) Stereoscopic representation showing the conformation of D-amino acid observed in the pocket. (**B**) The adenosine moiety and the carbonyl oxygen are tightly fixed. The only allowed flexibility would be the torsion around Cα-C' bond. This rotation gives rise to three theoretical possibilities of binding an L-amino acid. (**C**) Conformation I: the side chain swaps positions with Hα, severe short contacts of the side chain atoms including Cβ with active site residues can be seen. (**D**) Conformation II: the side chain swaps positions with NH₂ group, short contact of side chain with C75 of tRNA, also Cβ is 2.56 Å from amide nitrogen (N8) of the substrate. (**E**) Conformation III: the NH₂ group swaps positions with Hα, non-polar side chain of Pro150 provides unfavorable environment for NH₂ group.
The following figure supplements are available for figure 4:

**Figure supplement 1**. Theoretically possible modes of D-amino acid binding.

nucleotide (C75). In fact, the Cβ itself would have a short contact (2.56 Å) with the amide nitrogen of the substrate (ester oxygen in the real substrate). It should be highlighted here that the side chain rejection in both positions occurs at the Cβ level itself, which implies that an amino acid with even a minimal side chain like L-Ala will be rejected from occupying these two positions. In the third possibility of conformation III, the amino group would swap its position with Hα (*Figure 4E*). In this case, in addition to losing its hydrogen bonding interaction with Gly149 carbonyl oxygen, the amino group would be placed also in an unfavorable environment at a distance of 3.07 Å from the Cδ atom of the non-polar side chain of Pro150 (*Figure 4E*). This provides an elegant mechanistic design for L-chirality rejection from this pocket irrespective of the conformation and side chain chemistry of the incoming substrate. The rejection mechanism also rules out any other possible mode of D-aa binding than the one observed where the side chain is kept protruding out (*Figure 4—figure supplement 1*).

The 'cross-subunit' Gly-*cis*Pro motif plays a central role in the rejection of L-aas from binding in the pocket. The *cis* conformation of Pro150 is the key to ensuring that it cradles the chiral centre thus preventing both the amino group and the Cβ from occupying the position of Hα (*Figure 5A*). To facilitate this rejection mechanism, Pro150 side chain is positioned rigidly in *cis* conformation by a conserved hydrophobic base formed by Phe40, Val86, Ile143, and the DTD-specific invariant Met141 (*Figure 5—figure supplement 1*). The Gly149 and Pro150 carbonyl oxygens make H-bond interactions with the α-amino group and the Cβ of the substrate respectively, thereby reinforcing the binding of D-aa in the pocket. Both the carbonyl oxygens are also positioned tightly by cross-subunit interactions with Met141 main chain nitrogen and Gln88 side chain nitrogen, respectively (*Figure 5A*). The structure, therefore, suggests a strict rejection of L-aas from the pocket, enabling DTD to specifically remove only D-aas coupled to tRNAs.

## Conservation of the strict configuration specificity across species

In order to prove the strict rejection of L-aa by the active site of DTD, biochemical analyses with *Pf*DTD were performed. Although significant deacylation activity against D-Tyr-tRNA$^{Tyr}$ was observed at 500 pM *Pf*DTD, no L-Tyr-tRNA$^{Tyr}$ deacylation was found even with 1000-fold higher enzyme concentration at 500 nM (*Figure 5B*). Furthermore, to rule out the possibility of any *Plasmodium*-specific phenomenon and to test the universal nature of the rejection mechanism, we carried out deacylation experiments with *Ec*DTD as well. Similar to *Pf*DTD, *Ec*DTD showed significant deacylation of D-Tyr-tRNA$^{Tyr}$ with 50 nM enzyme, whereas no detectable L-Tyr-tRNA$^{Tyr}$ deacylation was seen even at 5 μM (*Figure 5C*). Biochemical studies with both enzymes not only confirm the stringent chiral specificity of this key process but also suggest conservation of the mechanism across species.

## Strict rejection of L-aa-tRNA as seen with NMR-based binding studies

We further probed the enantiomeric rejection mechanism in solution using NMR-based 2D $^{15}$N-$^1$H Transverse Relaxation Optimized Spectroscopy (TROSY) experiments with a nonhydrolyzable analog mimicking L-Tyr attached to tRNA$^{Tyr}$, L-Tyr3AA, and compared it with D-Tyr3AA. Titration of $^{15}$N-*Pf*DTD with D-Tyr3AA at molar ratios of 1:0, 1:5, 1:10, and 1:15 led to chemical shift perturbations in a number of resonances and showed saturation around 1:15, thereby clearly indicating a specific binding to *Pf*DTD (*Figure 5D*, *Figure 5—figure supplement 2*). On the other hand, L-Tyr3AA titration did not cause any change in the amide resonances of $^{15}$N-*Pf*DTD even up to 1:15 molar ratio, highlighting a complete lack of specific binding (*Figure 5D*, *Figure 5—figure supplement 2*). Thus, the 2D $^{15}$N-$^1$H TROSY studies further confirmed the strict rejection of L-aa from the active site of DTD.

## 2′-vs 3′- deacylase

Another important mechanistic aspect that is clearly evident from this structure is that DTD acts exclusively on D-aas charged on 3′-OH of the terminal adenosine. aaRSs aminoacylate tRNAs at either 2′-OH or 3′-OH in a class-dependent way (*Eriani et al., 1990*). Biochemical studies have revealed deacylation mechanism of DTD against aa–tRNA pairs belonging to both classes of aaRS. However, it was not clear whether DTDs would act on D-aas linked to 2′-OH or 3′-OH or both. The structure shows that the 2′-OH is positioned in a confined area with the help of tight interactions with Gly138 main chain nitrogen and Ser87 side chain hydroxyl group. Modeling even the simplest of amino acids on the 2′-OH shows severe steric clashes irrespective of the ribose pucker (*Figure 6A–C*). We have further confirmed this mechanistic proposal using 2D $^{15}$N-$^1$H TROSY experiments. Titration of $^{15}$N-*Pf*DTD with D-Tyr3AA showed chemical shift perturbations for a number of resonances (*Figure 5D*, *Figure 5—figure supplement 2*). On the

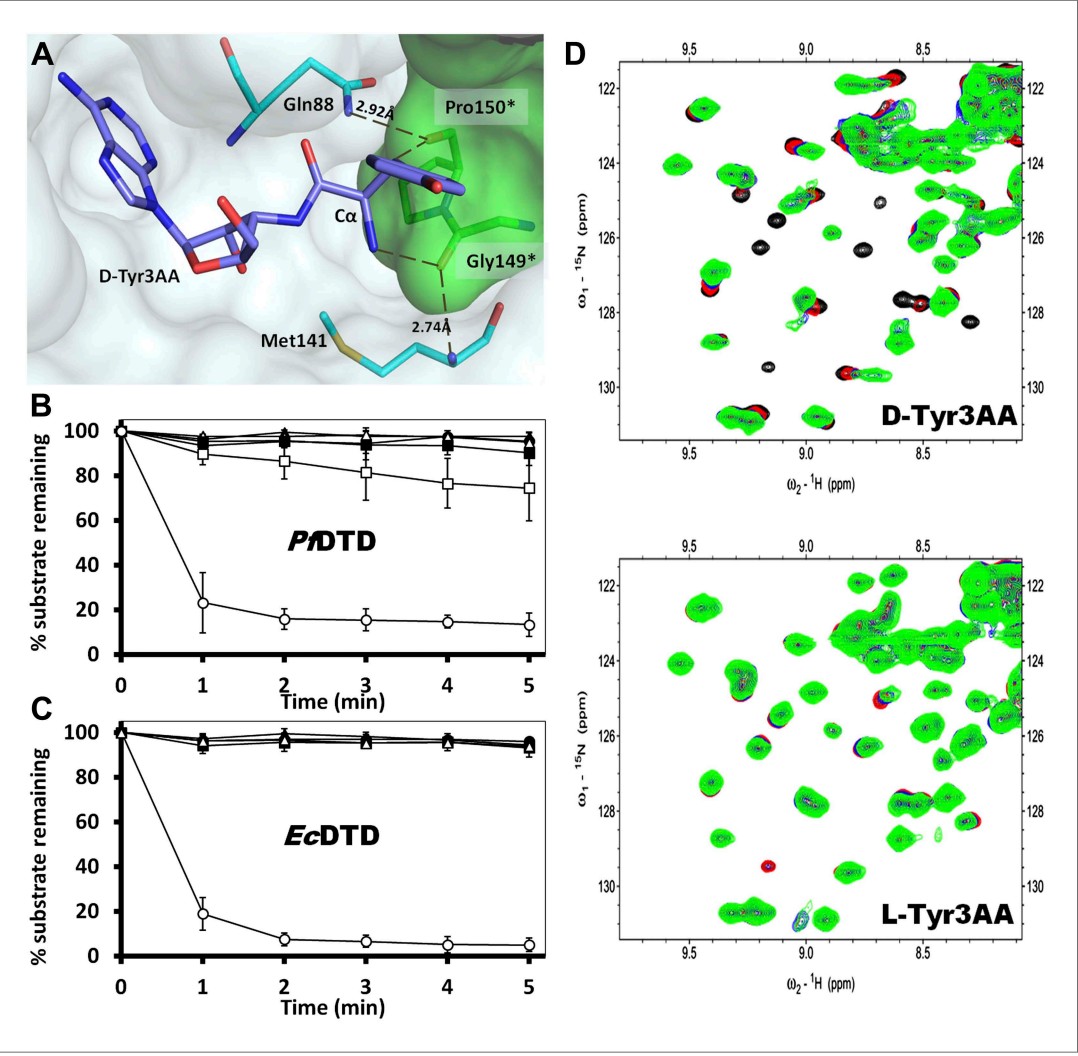

**Figure 5**. Strict configurational specificity of DTD. (**A**) The Gly-*cis*Pro motif from one monomer protrudes into the active site of the other monomer and cradles the chiral center of the substrate and provides basis for configuration selection. The carbonyl oxygens are tightly positioned by cross-subunit interactions. (**B**) Deacylation of L-Tyr-tRNA$^{Tyr}$ by buffer (——●——), 500 pM (——▲——), 5 nM (——△——), 50 nM (——■——), 500 nM (——□——) *Pf*DTD and D-Tyr-tRNA$^{Tyr}$ deacylation by 500 pM *Pf*DTD (——○——). (**C**) L-Tyr-tRNA$^{Tyr}$ deacylation by buffer (——●——), 50 nM (——▲——), 500 nM (——△——), 5 µM (——■——) *Ec*DTD and D-Tyr-tRNA$^{Tyr}$ deacylation by 50 nM *Ec*DTD (——○——). (**D**) Excerpts of overlay of 2D $^{15}$N-$^1$H TROSY obtained with 0.2 mM *Pf*DTD (black) and upon addition of 1 mM (red), 2 mM (blue), 3 mM (green) D-Tyr3AA and L-Tyr3AA.

The following figure supplements are available for figure 5:

**Figure supplement 1**. Hydrophobic base for Pro150 side chain.

**Figure supplement 2**. 2D $^{15}$N-$^1$H TROSY of *Pf*DTD with D-Tyr3AA and L-Tyr3AA.

**Figure supplement 3**. TLC-based deacylation assay with *Pf*DTD wild type against D-Tyr-tRNA$^{Tyr}$ and L-Tyr-tRNA$^{Tyr}$.

other hand, titration with D-Tyr2AA (analog of D-tyrosine bound to 2′-OH of adenosine) resulted in no observable chemical shift perturbations (**Figure 6D,E**). This confirms that the enzyme acts on tRNAs only when the amino acid is either attached to 3′-OH or transferred to 3′-OH from 2′-OH through rapid transesterification. A similar mechanistic mode of operation of Pab-NTD delineates this DTD-like fold as a 3′-specific deacylase enzyme (**Hussain et al., 2006, 2010**).

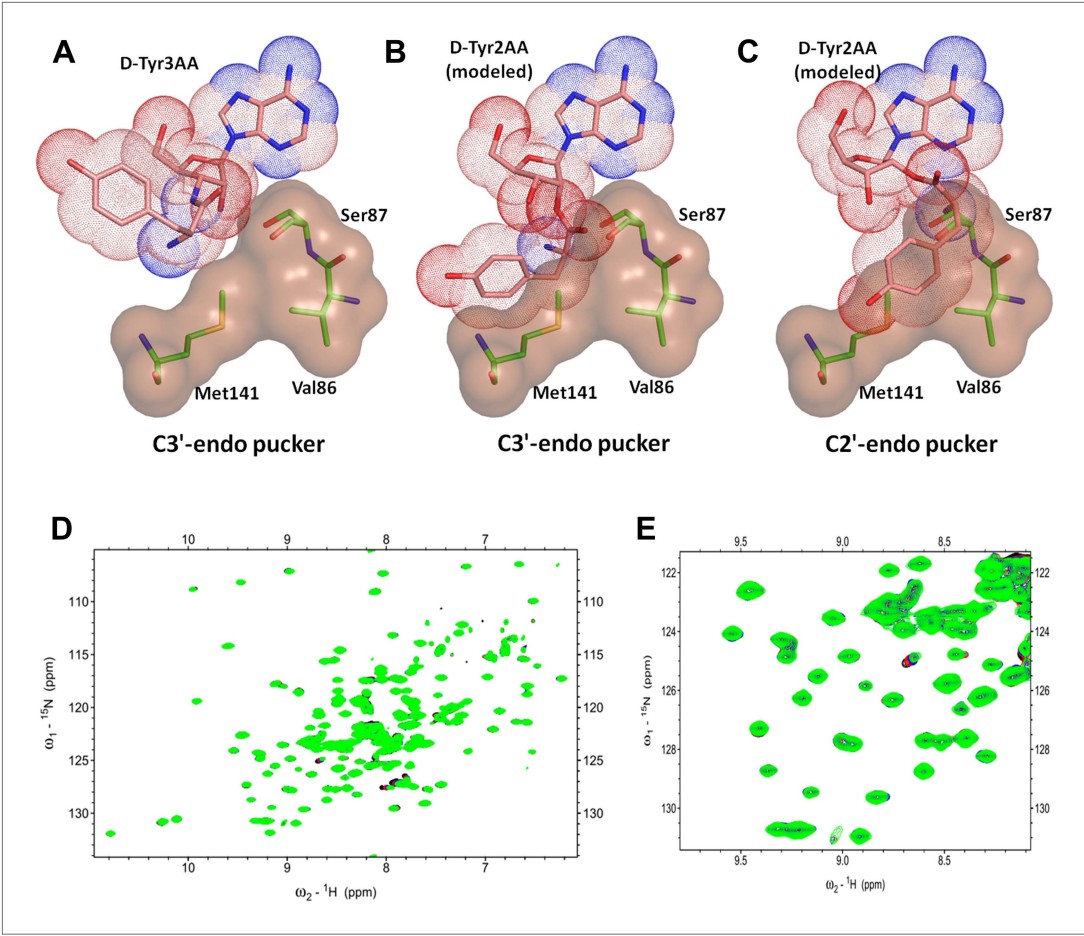

**Figure 6**. DTD is a strict 3'-specific deacylase. (**A**) Ribose moiety of D-Tyr3AA adopts C3'-endo pucker in the structure. Modeling the aminoacyl group on the 2'-OH shows serious steric clashes in (**B**) C3'-endo as well as (**C**) C2'-endo puckers. (**D**) Overlay of 2D $^{15}$N-$^1$H TROSY obtained with 0.2 mM *Pf*DTD (black) and upon addition of 1 mM (red), 2 mM (blue), 3 mM (green) D-Tyr2AA. (**E**) Excerpt of the overlay for clarity.

## Gly-*cis*Pro motif is essential for function

The mechanistic understanding based on the cognate substrate analog-bound structure suggests a crucial role for the cross-subunit Gly-*cis*Pro motif in enantioselectivity and rejection of L-aas from the pocket. To experimentally demonstrate the crucial role of this unique motif for DTD function, we carried out deacylation assays with *Pf*DTD by mutating these two critical residues. A complete loss of activity was observed for both G149A and P150A mutants (*Figure 7A*). We also carried out deacylation assay with G149A/P150A double mutant and similar to both single mutants, it showed a total loss of activity (*Figure 7A*). The biochemical studies, thus clearly, show that Gly-*cis*Pro motif is essential for DTD function. We further wanted to ensure that the observation is not *Plasmodium*-specific. Therefore, we performed the same biochemical study with the mutants of *Ec*DTD to ensure that the critical role of the Gly-*cis*Pro motif is universal. Similar to *Pf*DTD, both G137A and P138A mutants of *Ec*DTD showed a complete loss of deacylation function (*Figure 7B*). We also tested G137A/P138A double mutant for deacylation function and it also showed no activity like the individual point mutants (*Figure 7B*). The biochemical analyses with the mutants of both *Pf*DTD and *Ec*DTD prove the critical role played by the unique Gly-*cis*Pro motif in DTD function and also suggests the universality of its crucial role irrespective of the organism.

## Discussion

The study provides insights into a fundamental enantioselective mechanism involved in enforcement of homochirality in proteins by specifically decoupling D-aas from tRNAs. The earlier structural studies

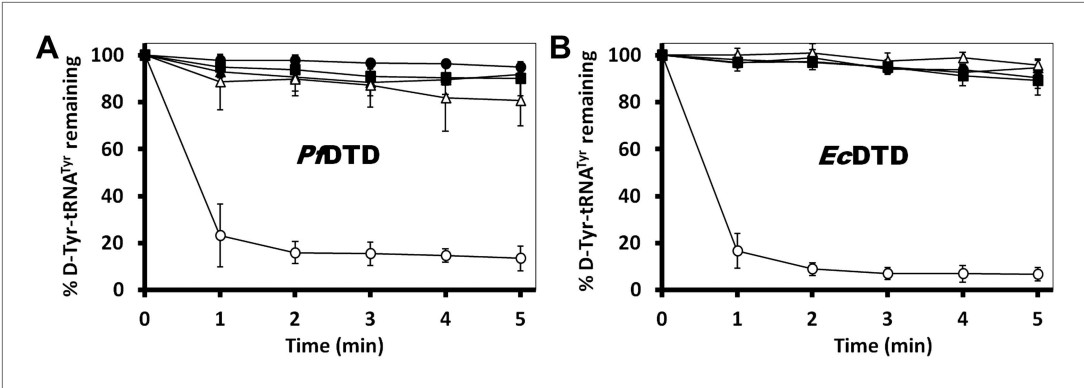

**Figure 7**. Critical role of Gly-*cis*Pro motif for DTD function. (**A**) Deacylation of D-Tyr-tRNA[Tyr] by buffer (━●━), *Pf*DTD wild type (━○━), G149A (━▲━), P150A (━△━) and G149A/P150A double mutant (━■━). 500 pM enzyme concentration was used for all reactions. (**B**) D-Tyr-tRNA[Tyr] deacylation by buffer (━●━), *Ec*DTD wild type (━○━), G137A (━▲━), P138A (━△━), and G137A/P138A double mutant (━■━). 50 nM enzyme concentration was used for all reactions.

The following figure supplements are available for figure 7:

**Figure supplement 1**. Circular dichroism spectra showing comparison of *Ec*DTD mutants with the wild type.

on DTD provided a mechanistic model based either on docking approaches using apo structure (**Lim et al., 2003**) or complex structures with ligands that do not mimic the cognate substrate (**Bhatt et al., 2010**). A superposition of the earlier known structures with that of the D-Tyr3AA-bound complex presented here shows that the docked substrate as well as the free D-aas and ADP were positioned outside the actual binding pocket (**Figure 1—figure supplement 1**). Therefore, the key to identifying the mechanism, as seen from this study, is the capturing of the D-Tyr3AA ligand that is bound in the actual substrate-binding pocket.

An analysis of all known structures of proofreading domains in complex with post-transfer substrate analogs helped us to define certain parameters such as percentage buried surface area of the ligand, number of interactions, conservation of interacting residues etc that can be used to assess the binding characteristics of ligand complexes (**Figure 1—figure supplement 2**, **Supplementary file 1C**). Comparison of these parameters from all known complex structures of proofreading domains with the structure presented in the current study places our structure in the same bracket as the other well-studied proofreading domains (**Supplementary file 1C**). We have mutated Phe89 that has been shown to stack with adenine in ADP-complex (**Bhatt et al., 2010**), to Ala and show that the mutant is as active as the wild-type *Pf*DTD (**Figure 3—figure supplement 2A,C**). The corresponding mutant F79A in *Ec*DTD was also completely active suggesting that the Phe has no significant role in binding the adenine (**Figure 3—figure supplement 2D**). Furthermore, the earlier work had implicated a conserved Thr90 as the catalytic residue that was proposed to mount a nucleophilic attack on the carbonyl carbon of the substrate (**Lim et al., 2003**; **Bhatt et al., 2010**). However, our analysis clearly shows that not only the distance (5.72 Å) of γ-hydroxyl group of Thr90 from the carbonyl carbon is unfavorable for any nucleophilic attack but also it is oriented away from the point of attack where it is strongly tethered to Thr152 main chain atoms through a highly conserved cross-subunit interaction (**Figure 3—figure supplement 2B**). To experimentally demonstrate that Thr90 is not the catalytic residue as had been proposed earlier, we mutated this residue to Ala in both *Pf*DTD and *Ec*DTD, and showed that they still efficiently deacylated D-Tyr-tRNA[Tyr] (**Figure 3—figure supplement 2C,D**). These data, therefore, rule out the earlier propositions not only with respect to the adenosine-binding site but also the catalytic mechanism.

More importantly, the current study identifies the key role of an invariant cross-subunit Gly-*cis*Pro motif in solving a fundamental problem of absolute configuration-based selectivity. The most striking feature of the Gly-*cis*Pro motif is the near-parallel fixation of the two carbonyl groups at an angle of ~20°, a highly conserved structural feature in DTDs irrespective of the presence or absence of ligand as seen in 72 different observations (including 10 from this study) from five different organisms (**Figure 8A**). The Ramachandran dihedral angles of both residues remarkably illustrate a striking conservation,

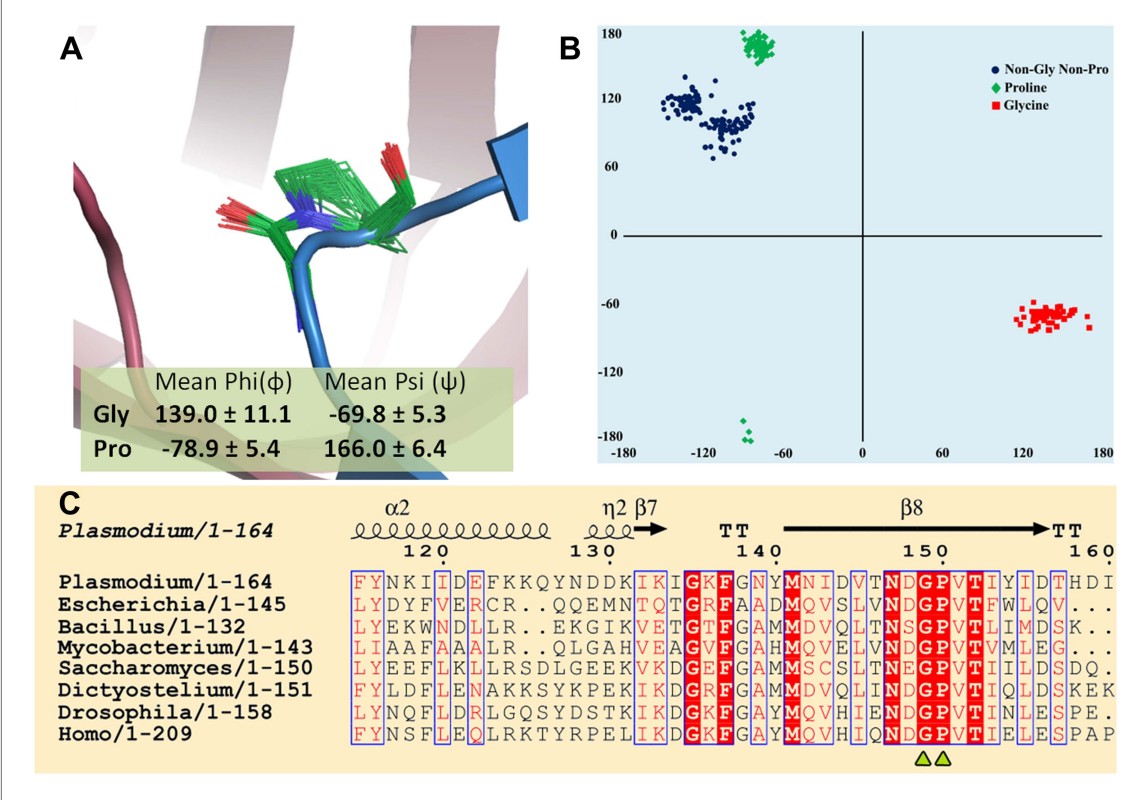

**Figure 8**. Strict conservation of the Gly-*cis*Pro motif in DTDs. (**A**) The structural superimposition of –NXGP(V/F)T– motif from 72 different monomers of DTD (10 from this study and 62 from PDB including DTDs from *E. coli*, *H. influenzae*, *Plasmodium falciparum*, *Aquifex aeolicus*, and *H. sapiens*) shows that the rigid fixation of Gly149 and Pro150 carbonyl groups is structurally conserved in all DTDs. (**B**) Ramachandran map for residues from –NXGP(V/F)T– motif from all DTD structures shows that glycine invariably occupies the lower right quadrant. (**C**) Both Gly and Pro are invariant in all DTD sequences.

which allows DTD to selectively recognize the chiral centre. It also provides a structural explanation for having an invariant Gly in that position as no other residue can normally lie in that region of Ramachandran map (*Figure 8B,C*). Since the cellular milieu will be in abundance with L-aa-tRNAs, when compared to D-aa-tRNAs, such a positioning of the critical enantioselective components, as seen here, prevents even a promiscuous deacylation of L-aa-tRNAs leading to their depletion from the pool, as shown by the biochemical studies with 1000-fold excess of DTD in two different systems. The essential role of Gly-*cis*Pro motif in chiral discrimination is also strongly indicated by its absolute invariance in all DTD sequences from eubacteria to higher eukaryotes (*Figure 8C*). Previous work has shown the ability of L-proline to catalyze asymmetric synthesis of simple sugars leading to their enantioenrichment (*Breslow and Cheng, 2010*; *Hein and Blackmond, 2012*). Based on the work, there has been a proposal of a role of L-proline in symmetry-breaking during the prebiotic era. In the present work also, we show the critical role of a proline residue as a part of a motif in a process involved in enforcement of homochirality.

Overall, the work has unveiled a fundamental cellular mechanism that is responsible for enforcing and perpetuating L-aa homochirality in proteins. A mechanistically unique solution to the problem of enantioselectivity employing two carbonyl oxygens from a 'cross-subunit' Gly-*cis*Pro dipeptide has been shown to be responsible for D-chirality selection and strict L-chirality rejection from the active site of DTD. The conserved and indispensable nature of the motif in DTD argues strongly for its crucial role in solving this key chiral discrimination problem in biology. The presence of DTD-fold and function in all kingdoms of life suggests an important role such systems have played in enforcing homochirality during early evolution of the translational apparatus, and high levels of expression in neuronal cells indicate a crucial role of DTD in higher organisms, which still needs to be explored.

## Materials and methods

### Cloning, expression and protein purification

The gene encoding DTD was PCR amplified from *P. falciparum* genomic DNA and inserted between *Nde*I and *Xho*I sites of pET-21b vector (Novagen, Billerica, MA). For untagged construct, a stop codon was incorporated in the reverse primer whereas in case of C-terminal 6X His-tagged (C-His) construct, there was no stop codon in the reverse primer. Untagged protein was used for crystallization and biochemical analysis, while NMR experiments were performed with C-His protein. The recombinant plasmid containing our gene of interest was transformed in *E. coli* BL21 (DE3) cells for overexpression. The untagged protein was purified by a two-step protocol including cation exchange chromatography (CEC) followed by gel filtration chromatography (GFC). In CEC, the induced cell lysate was loaded onto Sulfopropyl-Sepharose column (Amersham Pharmacia, UK) pre-equilibrated with 50 mM BisTris pH 6.5, 20 mM NaCl and then eluted in a linear gradient of NaCl from 20 mM to 500 mM. The eluted protein was further purified to homogeneity by GFC using a Superdex-75 column (Amersham Pharmacia). The final protein was concentrated to 10 mg/ml. *Ec*DTD was purified as mentioned previously (*Hussain et al., 2006*). All proteins were expressed normally except for G137A and double mutant G137A/P138A of *Ec*DTD, which were purified from inclusion bodies using the following procedure. After lysis, the inclusion bodies were washed thoroughly with buffer containing 1% Triton X-100, followed by 1% sodium deoxycholate wash and finally incubated overnight in unfolding buffer containing 6M guanidinium hydrochloride (GdmHCl). The unfolded protein was then loaded onto Ni-NTA column (Amersham Pharmacia) pre-equilibrated with unfolding buffer and subsequently washed with 1% Triton X-100, followed by 0.1% β-cyclodextrin wash. This was followed by 30 mM imidazole wash to get rid of any contaminant proteins. The protein was finally eluted with 250 mM imidazole and immediately diluted in refolding buffer containing 400 mM L-Arg. The protein was further purified to homogeneity using GFC. Circular Dichroism analysis was performed to ensure that the proteins were properly folded (*Figure 7—figure supplement 1*).

### Co-crystallization with substrate-mimicking analog D-Tyr3AA

Co-crystallization was attempted with a number of constructs of DTD from *E. coli*, *Mycobacterium tuberculosis*, *Vibrio cholera*, *Leishmania major* but none of them yielded a ligand-bound structure. Successful co-crystallization was achieved only with *Pf*DTD. The nonhydrolyzable analogs D-Tyr3AA, L-Tyr3AA, and D-Tyr2AA were obtained after custom synthesis from Jena Biosciences, Germany. The pure protein sample was mixed with the ligand in a molar ratio of 1:20 and the premix was incubated at 4°C overnight. Initial crystallization conditions were screened at 4°C and 20°C with Index and Crystal screen 1 and 2 (Hampton Research, Aliso Viejo, CA) and JBS classic (Jena Biosciences) in sitting drop setups using 96-well plates from MRC. The experiments were set up by mixing 1 µl of protein:ligand premix with 1 µl of reservoir buffer with the help of Mosquito crystallization robot (TTP LabTech, UK). The hits obtained were further optimized in a hanging drop vapor diffusion setup using 24-well Iwaki plates. *Pf*DTD+D-Tyr3AA crystal I was obtained in 0.1M HEPES pH 7.0, 0.6M NaCl, 32% PEG3350, while crystal II of the same was obtained in 0.1M BisTris pH 6.0, 0.4 M NaCl, 28% PEG3350.

### X-ray diffraction data collection and structure determination

The diffraction data were collected at the in-house X-ray facility after screening several hundreds of ligand complex crystals to get high resolution datasets. The dataset for *Pf*DTD+D-Tyr3AA crystal I was collected using RigakuMicromax007 HF rotating-anode generator that produces CuKα X-rays of wavelength 1.54 Å and MAR345dtb image-plate detector from MAR Research. The crystal was mounted on a nylon loop and flash-cooled directly without the use of any cryoprotectant solution in a nitrogen-gas stream at 100 K using Oxford Cryostreamcooler (Oxford Cryosystems, UK). The dataset for *Pf*DTD+D-Tyr3AA crystal II was collected using FR-E+ SuperBright X-ray generator from Rigaku equipped with VariMax HF optic and R-AXIS IV++ image plate detector. The data were processed using HKL2000 (*Otwinowski and Minor, 1997*) and the structure was solved by molecular replacement using MOLREP-AUTO MR from the CCP4 suite (*CCP4, 1994*) with *Pf*DTD apo structure (PDB id: 3KNF) as the search model. The structure was refined with the help of CNS (*Brunger et al., 1998*) and REFMAC (*Murshudov et al., 1997*), while COOT (*Emsley and Cowtan, 2004*) was used for model building. The restraints for refinement of ligand molecules were obtained from PRODRG server

(*Schuttelkopf and van Aalten, 2004*). The structure was validated using PROCHECK (*Laskowski et al., 1993*) and the figures were generated with the help of PyMOL (*Schrodinger, 2010*).

## Biochemical assays

The mutants for biochemical assays were generated using QuickChange XL site-directed kit (Stratagene, La Jolla, CA) and the proteins were purified by the same protocol as for the wild type. *E. coli* tRNA$^{Tyr}$ was transcribed in vitro using MEGAshortscript (Ambion, Austin, TX) and 3' end-labeled using standard protocol by incubating the tRNA with CCA-adding enzyme in presence of [α-$^{32}$P]-ATP (*Ledoux and Uhlenbeck, 2008*). D-Tyr-tRNA$^{Tyr}$ and L-Tyr-tRNA$^{Tyr}$ were generated by incubating 20 mM Tris pH 7.8, 7 mM MgCl$_2$, 5 mM Dithiothreitol (DTT), 2 mM ATP, 0.2 mM amino acid (D-Tyr or L-Tyr), 0.5 µM labeled tRNA$^{Tyr}$, 1 U/ml pyrophosphatase with 2 µM purified *E. coli* TyrRS at 37°C for 15 min. Aminoacylation reaction was followed by phenol extraction and ethanol precipitation of aminoacylated tRNA, which was finally resuspended in 5 mM sodium acetate pH 4.6. Deacylation assays were performed by incubating 20 mM Tris pH 7.2, 5 mM MgCl$_2$, 5 mM DTT, 0.2 mg/ml bovine serum albumin (BSA), 0.2 µM labeled D-Tyr-tRNA$^{Tyr}$ or L-Tyr-tRNA$^{Tyr}$ at 30°C with 500 pM of *Pf*DTD and 50 nM of *Ec*DTD or the mutants enzyme as the case may be. Reaction mix at various time points were subjected to S1 nuclease digestion for 30 min at 22°C and analyzed by thin-layer chromatography (TLC) by spotting 1 µl on PEI cellulose sheet (Merck KGaA, Germany). An example of a TLC run has been shown in *Figure 5—figure supplement 3*. The mobile phase for TLC was composed of 100 mM ammonium chloride and 5% glacial acetic acid. TLC sheets were exposed to imaging plate from Fujifilm, Japan. Phosphor imaging was done using Typhoon Trio Variable Mode Imager (Amersham Biosciences, Piscataway, NJ) and Image Gauge V4.0 software was used for quantification. Each experiment was carried out in triplicates.

## Transverse relaxation optimized NMR spectroscopy

2D $^{15}$N-$^1$H TROSY experiments were performed on a Bruker 600 MHz NMR spectrometer equipped with triple resonance cryoprobe (Bruker, Billerica, MA). C-His construct of *Pf*DTD was expressed in minimal media with $^{15}$NH$_4$Cl as the sole nitrogen source in order to achieve uniform labeling. The protein was purified by affinity chromatography using Ni-NTA column in batch mode. For binding studies, 200 µM U-$^{15}$N-*Pf*DTD in 50 mM HEPES pH 7.0, 50 mM NaCl was titrated with substrate analogs. Chemical shift perturbations in *Pf*DTD upon titration were monitored by a series of 2D $^{15}$N-$^1$H TROSY spectra collected with increasing concentrations of ligand. Four datasets were recorded for each ligand at protein:ligand molar ratios of 1:0, 1:5, 1:10, and 1:15. The experiments were repeated twice with two different batches of protein. The data processing and figure preparation were done using Sparky.

## Acknowledgements

SA and SBR thank Council of Scientific and Industrial Research (CSIR), India for funding. RS acknowledges funding from Swarnajayanti Fellowship of Department of Science and Technology, India and 12th Five Year Plan Project BSC0113 of CSIR, India.

## Additional information

### Funding

| Funder | Grant reference number | Author |
| --- | --- | --- |
| Swarnajayanti Fellowship, DST India | | Rajan Sankaranarayanan |
| CSIR India | BSC0113 | Rajan Sankaranarayanan |

The funders had no role in study design, data collection and interpretation, or the decision to submit the work for publication.

### Author contributions

SA, SBR, Acquisition of data, Analysis and interpretation of data, Drafting or revising the article; VK, JC, SM, TH, SPK, MVD, Acquisition of data, Analysis and interpretation of data; RS, Conception and design, Analysis and interpretation of data, Drafting or revising the article

# Additional files

## Supplementary files

• Supplementary file 1. (A) Interaction distances (Å) of the aminoacyl moiety of D-Tyr3AA with active site residues; (B) Atomic B-factors (Å$^2$) of D-Tyr3AA atoms in all monomers from crystal forms I and II; (C) Various aspects of post-transfer substrate analog-bound structures of proofreading domains from earlier work and the current study along with ADP-bound structure of PfDTD (*Bhatt et al., 2010*) and docked substrate model (*Lim et al., 2003*).

## Major datasets

The following datasets were generated:

| Author(s) | Year | Dataset title | Dataset ID and/or URL | Database, license, and accessibility information |
| --- | --- | --- | --- | --- |
| Ahmad S, Routh SB, Kamarthapu V, Sankaranarayanan R | 2013 | D-aminoacyl-tRNA deacylase (DTD) from *Plasmodium falciparum* in complex with D-tyrosyl-3'-aminoadenosine at 1.86 Angstrom resolution | http://www.rcsb.org/pdb/explore/explore.do?structureId=4NBI | Publicly available at RCSB Protein Data Bank (http://www.rcsb.org). |
| Ahmad S, Routh SB, Kamarthapu V, Sankaranarayanan R | 2013 | D-aminoacyl-tRNA deacylase (DTD) from *Plasmodium falciparum* in complex with D-tyrosyl-3'-aminoadenosine at 2.20 Angstrom resolution | http://www.rcsb.org/pdb/explore/explore.do?structureId=4NBJ | Publicly available at RCSB Protein Data Bank (http://www.rcsb.org). |

The following previously published datasets were used:

| Author(s) | Year | Dataset title | Dataset ID and/or URL | Database, license, and accessibility information |
| --- | --- | --- | --- | --- |
| Bhatt TK, Yogavel M, Wydau S, Berwal R, Sharma A | 2010 | Ligand-bound structures provide atomic snapshots for the catalytic mechanism of D-amino acid deacylase | http://www.rcsb.org/pdb/explore/explore.do?structureId=3knf | Publicly available at RCSB Protein Data Bank (http://www.rcsb.org). |
| Bhatt TK, Yogavel M, Wydau S, Berwal R, Sharma A | 2010 | Ligand-bound structures provide atomic snapshots for the catalytic mechanism of D-amino acid deacylase | http://www.rcsb.org/pdb/explore/explore.do?structureId=3ko5 | Publicly available at RCSB Protein Data Bank (http://www.rcsb.org). |

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
