## [Decision Letter]

[Editors’ note: the authors performed additional work to address the concerns raised in the first round of peer review and submitted for further consideration. The two decision letters after peer review are shown below.]

Thank you for choosing to send your work entitled “Mechanism of chiral proofreading during translation of the genetic code” for consideration at *eLife*. Your full submission has been evaluated by a Senior editor and 2 peer reviewers, and the decision was reached after discussions between the reviewers. We regret to inform you that your work will not be considered further for publication.

In this paper, the X-ray crystal structure of *Plasmodium falciparum* D-aminoacyl-tRNA deacylase (DTD) in a liganded complex with a non-hydrolysable D-tyrosine analogue, D-tyrosyl-3'-aminoadenosine, is presented. The reviewers recognize that the work is a significant contribution to the field and provides further insight into the catalytic mechanism of this enzyme, the details of which could not be determined from the previously solved apo-structure. The authors use mutagenic studies to demonstrate the crucial catalytic role of the highly conserved Gly-*cis*Pro motif within the enzyme as well as showing structurally the importance of this motif in chiral selection. In addition the findings also indicate that this particular DTD is likely to only act upon aminoacylated-tRNA species when the amino acid is attached to the 3' hydroxyl of the terminal adenine of the tRNA donor molecule. Nevertheless, the review has raised some major concerns that have led to the conclusion that the paper is not suitable for publication by *eLife*.

1) The major issues arise from placing the present work in the context of what is already known about the enzyme. The crystal structure of DTD has been determined earlier. A model docking the substrate analogs with DTD from different sources have also been obtained. Crystal structures of PfDTD with adenosine and different D amino acids have led to a mechanical model that is contrary to the model suggested here. Conclusions in the present manuscript go too far beyond what the data indicate.

The details of the reaction mechanism are still not completely defined, and so one has to make the hypothesis that the observed binding mode is the relevant one. The statement ‘Previous attempts using several apo and complex structures ... yielded the first cognate ligand-bound structure and provided the structural basis of this fundamental cellular process’ is misleading. It has been shown that D-Tyr-adenosine was not hydrolyzed by DTD, but a D-Tyr-esterified oligonucleotide produced by RNase T1 digestion of D-Tyr-tRNA having a 19mer oligonucleotide, is its substrate (16). Therefore, present ligand is also a model system for interpretation of the possible mode of substrate binding and catalytic action.

2) In addition to Gly*cis*Pro, there are several other amino acids involved in the interactions with the ligand (Figure 1) and if the true substrate that is much larger were to be bound, there would have been many contacts. The authors spent a lot of effort in highlighting the conserved nature of these two residues and their stereochemistry. How does one rule out other possible conserved interactions even if there are sequence differences in the binding pocket?

3) The binding pocket conformation has been assumed to be very rigid while attempting to illustrate that L-amino acid analogs would be sterically excluded (Figure 2). This has been substantially based on the conserved nature of the Gly*cis*Pro motif (Figure 6). In principle, the substrate binding pocket of an enzyme has to have certain amount of plasticity and even if the stereochemistry of a couple of residues is conserved, possible structural changes due to the plasticity associated with other residues can not be ruled out. Indeed, this argument is consistent with the degenerate recognition of diverse D-amino acids in case of the same enzyme (3). This is important particularly considering that the true substrate is much bigger than the ligand used here. While it is accepted that the L-amino acids are rejected, one has to be cautious about interpreting the mechanism based on the rigidity of a motif consisting of only two amino acids from the binding pocket.

4) [3] have provided ‘atomic snap shots’ for the catalytic mechanism of DTD based on the crystal structures of several complexes of DTD with many D-amino acids and ADP binding. The model proposed in their case emphasizes substantial plasticity at the binding site and highlights the possible catalytic steps. The authors of the present study reject that model based on two counts: they were not cognate ligand complexes and their binding modes are different. Knowing that D-Tyr-adenosine is also not a true substrate of DTD, better explanation would be required to rule out the earlier described binding modes considering they also involve D-amino acids and adenosine.

[Editors’ note: what now follows is the decision letter after additional work had been performed.]

Thank you for submitting your work entitled “Mechanism of chiral proofreading during translation of the genetic code” for further consideration at *eLife*. Your revised article has been favorably evaluated by a Senior editor and the original two reviewers. The manuscript has been improved but there are some remaining issues that need to be addressed before acceptance, as outlined below:

In this paper, the X-ray crystal structure of *Plasmodium falciparum* D-aminoacyl-tRNA deacylase (DTD) in a liganded complex with a non-hydrolysable D-tyrosine analogue, D-tyrosyl-3'-aminoadenosine, is presented. The work is a significant contribution to the field and provides further insight into the catalytic mechanism of this enzyme, the details of which could not be determined from the previously solved apo-structure. The authors use mutagenic studies to demonstrate the crucial catalytic role of the highly conserved Gly-*cis*Pro motif within the enzyme as well as showing structurally the importance of this motif in chiral selection. In addition the findings also indicate that this particular DTD is likely to only act upon aminoacylated-tRNA species when the amino acid is attached to the 3' hydroxyl of the terminal adenine of the tRNA donor molecule.

An earlier version of this manuscript had been rejected previously, but the authors have done a good job of responding to the criticisms of the previous version by providing additional data concerning the validity of the binding mode of the substrate analog, and by making changes to the text of the manuscript.

The authors should revise the manuscript taking into account the following points. The paper will not be sent out for review again and the editor will make a final decision based the revisions made to the manuscript. Please respond to all of these points by revising the manuscript appropriately.

1) The authors now describe mutations that indicate that the binding mode they see is relevant to catalysis. They also carry out some mutations around the active site that support nucleotide-mediated catalysis, although this point in not rigorously proven because the data are negative (i.e., mutations do not affect catalysis substantially). Hence, they should be careful to tone down the discussion regarding previously solved structures of DTD with other ligands. The focus should be on what can be definitively concluded from the combination of structures presented here and other earlier structures rather than just over stating apparent ‘failures’ of either. This would allow for better discussion of the different catalytic mechanisms proposed from earlier structures obtained with other ligands and what is proposed to be the case here. Specifically, the Introduction should describe the earlier structural work and place the present work properly in context of the previous work, before moving on to a discussion of the present work.

2) It is very difficult to place the results discussed here in the context of DTD from various species (eukaryotes, prokaryotes, mammals...). The Abstract should state which species is analyzed, and the Introduction should also state more clearly the extent to which DTD is conserved in different species and which ones are analyzed here. It is true that this information is available in the manuscript, but it should be provided up front and in a way that the reader can immediately appreciate.

3) The word “novel” should be deleted from the Abstract. It is not clear in what sense the solution that this enzyme has reached for specificity is “novel” and in any case the novelty should be left for others to judge.

4) In the discussion of the active site mutants, it is stated “The above data clearly demonstrate that none of the protein residues around the scissile bond are involved in catalysis.” Since only negative data are presented, this statement is too strong. Change “clearly demonstrate” to “indicate” or some other appropriate word. Likewise, a little later the paper states “…the above data strongly suggest that the DTD fold is designed to be an RNA-based catalyst in the proofreading reaction”. Again, in the absence of positive data about the mechanism, replace “strongly suggest” by “are consistent with” or some other appropriate wording.

5) Towards the end of the paper it is said “These data, therefore, clearly show the lack of functional relevance of the binding modes proposed earlier...”. This is too strong. How do the authors know that the previously seen binding modes are not intermediates for some part of the process of catalysis? Remove this wording and replace with appropriate text.

6) A little later, it is stated “More importantly, the current study identifies and reveals for the first time the key role of an invariant cross-subunit Gly-*cis*Pro motif in solving....”. Delete “and reveals for the first time”. This work is built on earlier structural work and it is sufficient to say that it identifies a key role for this motif.

7) The paragraph in the Discussion beginning “The study opens up important questions....” should be deleted. It is too speculative, and it is far from certain that the structure can be used to design DTD enzymes to create a “D-amino acid world”.

---

## [Author Response]

*1) The details of the reaction mechanism are still not completely defined, and so one has to make the hypothesis that the observed binding mode is the relevant one. The statement ‘Previous attempts using several apo and complex structures ... yielded the first cognate ligand-bound structure and provided the structural basis of this fundamental cellular process’ is misleading. It has been shown that D-Tyr-adenosine was not hydrolyzed by DTD, but a D-Tyr-esterified oligonucleotide produced by RNase T1 digestion of D-Tyr-tRNA having a 19mer oligonucleotide, is its substrate (*[16]*). Therefore, present ligand is also a model system for interpretation of the possible mode of substrate binding and catalytic action*.

We totally agree with the reviewers that the ligand used in the study is also not the real substrate for DTD. However, such post transfer analogs where the amino acid is linked to the adenosine moiety through non-hydrolyzable amide linkage has been extensively used in crystallographic studies to elucidate proofreading mechanisms in atomic details. We have now provided in the main text all the references where these analogs were used to decipher the structural basis of proofreading mechanism both in class I and class II aminoacyl-tRNA synthetases, which were further validated biochemically. In our current manuscript, we have added additional data to experimentally demonstrate that the ligand-bound complex we have obtained is the functionally relevant one by disrupting the adenine pocket with the help of mutations and showing that it leads to complete loss of activity in two different systems, i.e., *Pf*DTD and *Ec*DTD (Figure 3—figure supplement 1). As can be seen, both these mutations were selected to have opposing structural consequence, one protrudes in the adenine binding pocket and the other removes the naturally occurring stacking interaction. These data clearly show that the adenine binding mode identified in our study is used for the biological function of DTD. In addition, as elaborated later, the mutation of the adenine binding site as predicted by Bhatt et al. did not alter the activity, again done in both *Pf*DTD and *Ec*DTD, clearly showing that it is not the functional adenosine pocket. However, as correctly pointed out by the reviewers, we have modified the text to avoid misleading statements. We have now explicitly referred to our ligand as ‘substrate-mimicking analog’ instead of ‘cognate ligand’.

Further, we have now addressed the mechanism of DTD in a more complete way and propose an RNA-assisted catalytic mechanism involving the role of 2'-OH. Our additional mutational data including 7 mutants in *Pf*DTD and 4 mutants in *Ec*DTD, where we mutated every residue in the vicinity of the scissile bond, clearly demonstrate that none of the protein side chains play any role in catalysis (Figure 3—figure supplement 1). Moreover, the role of 2'-OH in catalysis has also been suggested in the case of Pab-NTD, which shares a striking structural similarity with DTD. Although, the catalytic role of vicinal hydroxyl group of ribose in proofreading has been experimentally shown in case of Phenylalanyl-tRNA synthetase (PheRS) by using deoxy analogs (Ling et al., 2010, PNAS (104), 72-77), doing the same for DTD or Pab-NTD has some practical limitations. In the case of Threonyl-tRNA synthetase (ThrRS), 2'-OH has been shown to play a critical role in aminoacylation reaction also, which means that tRNA^Thr^ with a terminal 2'-deoxyadenosine (2'-dA) cannot be charged. In the case of Tyrosyl-tRNA synthetase (TyrRS), the problem is even worse since it charges the amino acid on the 2'-OH of tRNA which then gets transesterified to 3'-OH. Therefore, generating substrate for DTD with tRNA^Tyr^-2'-dA would not be possible. These practical limitations make it difficult to directly verify the catalytic role of 2'-OH in both DTD and Pab-NTD. However, the lack of involvement of any protein residues in catalysis strongly suggests a role of RNA in catalysis, as shown in the case of PheRS. The above aspect on catalysis with the new mutational data has been extensively dealt with in the new manuscript, while the overall focus is still on the unique chiral discrimination mechanism of DTD.

As we understand, one of the major concerns of the reviewers earlier was about the ‘authenticity’ of the ligand-binding mode as observed in our complex structure. Earlier we have presented mutational data on the Gly-*cis*Pro that abolished the biochemical activity of DTD. Now, we present direct experimental evidence, as correctly prompted by the reviewers, on the indispensible role of adenosine binding pocket as well. Therefore, independent mutational data on two discrete sites separated by ∼12 Å, as identified by D-Tyr3AA-bound complex, showing an essential role in activity clearly demonstrate that the binding mode observed is the functionally relevant one.

*2) In addition to Gly*cis*Pro, there are several other amino acids involved in the interactions with the ligand (*Figure 1*) and if the true substrate that is much larger were to be bound, there would have been many contacts. The authors spent a lot of effort in highlighting the conserved nature of these two residues and their stereochemistry. How does one rule out other possible conserved interactions even if there are sequence differences in the binding pocket*?

We absolutely agree with the reviewers that there are conserved critical interactions of the ligand with residues other than Gly-*cis*Pro such as invariant Phe137 and highly conserved Ser87, Gln88, and Gly138. We probably downplayed these interactions inadvertently because of our overemphasis on Gly-*cis*Pro motif and its critical role in enantioselection. We have now made suitable modifications in the manuscript to discuss more about the other conserved interactions of the ligand in DTD. We have added another section in the Results to bring out the essence of these interactions more. In addition, we have compared these interactions in DTD with that in Pab-NTD to highlight the conservation of adenine recognition in this fold (Figure 2—figure supplement 4). We have also provided more mutational data to support the critical role of these interactions in ligand binding (Figure 3).

*3) The binding pocket conformation has been assumed to be very rigid while attempting to illustrate that L-amino acid analogs would be sterically excluded (*Figure 2*). This has been substantially based on the conserved nature of the Gly* cis*Pro motif (*Figure 6*). In principle, the substrate binding pocket of an enzyme has to have certain amount of plasticity and even if the stereochemistry of a couple of residues is conserved, possible structural changes due to the plasticity associated with other residues can not be ruled out. Indeed, this argument is consistent with the degenerate recognition of diverse D-amino acids in case of the same enzyme (*[3]*). This is important particularly considering that the true substrate is much bigger than the ligand used here. While it is accepted that the L-amino acids are rejected, one has to be cautious about interpreting the mechanism based on the rigidity of a motif consisting of only two amino acids from the binding pocket*.

We thank the reviewers for pointing out this very important aspect that we again feel is an impression resulting from our overemphasis on Gly-*cis*Pro motif and its conserved stereochemistry. In reality, the active site of DTD has substantial plasticity as highlighted by the subtle rearrangements of the active site residues upon ligand binding. We have now included a figure that compares the active site residues in the apo form and the ligand-bound form to illustrate this plasticity associated with the active site (Figure 2). However, we respectfully disagree that the ‘degenerate recognition of diverse D-amino acids’, as observed by Bhatt et al., in any way indicates the plasticity of active site. We have dealt with the problems associated with these structures in detail in concern 4.

*4)*
[3]
*have provided ‘atomic snap shots’ for the catalytic mechanism of DTD based on the crystal structures of several complexes of DTD with many D-amino acids and ADP binding. The model proposed in their case emphasizes substantial plasticity at the binding site and highlights the possible catalytic steps. The authors of the present study reject that model based on two counts: they were not cognate ligand complexes and their binding modes are different. Knowing that D-Tyr-adenosine is also not a true substrate of DTD, better explanation would be required to rule out the earlier described binding modes considering they also involve D-amino acids and adenosine*.

We now provide confirmatory experimental evidence to not only rule out the earlier ligand binding modes but also the proposed catalytic mechanism. We have also included an analysis that compares the earlier binding modes with our complex structure in the light of all known post transfer substrate analog-bound structures of proofreading domains (Figure 1—figure supplement 2). Our analysis highlights features of the earlier binding modes that are totally uncharacteristic of biologically relevant complex. We deal with the ADP and D-amino acid complexes separately below:

ADP complex:

First of all, ADP is neither a substrate for DTD nor for any other proofreading domain. Even if we consider it as a mimic of the substrate since it also has an adenosine moiety, there are serious issues with the mode of binding observed in the ADP-bound structure. Going by the general principles of enzyme-substrate interactions, the ligand in a biologically relevant complex is expected to bind in the deepest available pocket with strong surface complementarity and chemical compatibility with a conserved network of interactions. As we now show in Figure 1—figure supplement 2, where we compare the ADP-bound structure with our structure and all the known complexes of proofreading domains, the ADP binds in a site that leaves the deepest pocket unoccupied unlike our structure or any other known complex structure of proofreading domains. Moreover, the striking surface complementarity that is evident in all complex structures including ours is totally absent in ADP complex. Secondly, the binding of ADP buries less than 50% of the ligand that is much less than the values that are mostly in the range of 80% for all known complex structures including ours. Thirdly, the biologically relevant complexes exhibit a remarkable redundancy of observation wherein the presence of ligand is observed in every monomer. Although such a redundancy is seen in our structure, where the ligand D-Tyr3AA has been observed in all ten monomers from 2 different crystal forms, ADP has only been sporadically observed in just 7 out of 18 monomers. Lastly and most importantly, a biologically relevant complex is expected to have a substantial number of interactions between the ligand, and the enzyme and the residues involved in these interactions must be evolutionarily conserved. Such evolutionarily conserved interactions are observed in all proofreading domain complexes as well as our structure but are missing in ADP complex.

In order to experimentally prove that the ADP complex has no biological relevance we have mutated Phe89 (which is the only conserved residue that interacts with ADP by stacking with the adenine ring) to Ala in both *Pf*DTD and *Ec*DTD, and show that the mutant is still completely active. To this effect we have added Figure 3—figure supplement 2 in the manuscript. These data, therefore, clearly rule out any functional significance of the ADP complex.

D-amino acid complexes:

Just as in case of ADP, free D-amino acids are also not the substrate for DTD rather they are the products of the reaction. Even if we assume them to partly mimic the actual substrate, the D-amino acid complexes reported have some serious problems as discussed below. First of all, these structures (except D-Lysine complex) have been solved at resolutions ranging from 2.75 Å to 3.00 Å, which makes the accurate modeling of free D-amino acids difficult. It may be noted that in most of these structures even waters have not been modeled. Secondly, in most cases the D-amino acid has only been modeled in just 1 out of 6 monomers highlighting a lack of redundancy of observations. Thirdly, the D-amino acids have been modeled with minimal interactions with the protein (in some cases there is no interaction between D-amino acid and protein at all). Even these minimal interactions with protein residues are not evolutionarily conserved at all. All these features are highly unexpected of a functionally relevant complex. Moreover, as we have illustrated in Figure 1—figure supplement 1, the position as well as orientation of all D-amino acids is highly variable. In some cases the carboxylate group points towards the enzyme, whereas in some cases it points outwards. This variable mode of D-amino acid binding has been possibly misconstrued as plasticity of active site and has been used to explain DTD’s activity against multiple D-amino acids (3). We do not think that the above argument holds good given the fact that the actual substrate for DTD is D-aminoacyl-tRNA, which means that D-amino acid would be ester linked to the terminal adenosine of tRNA. Therefore, the position as well as the orientation of D-amino acid in a functional complex would be fixed by the adenosine-binding pocket. In order to accommodate all the modes of D-amino acid binding observed, DTD must have multiple adenosine binding pockets as well, which is a highly improbable scenario. In any case, the observed modes of D-amino acid complexes do not provide any explanation for the chiral selection mechanism. On the other hand, the side chain-independent mode of D-Tyr3AA binding that we observe in the current study not only elucidates this chiral discrimination mechanism but also elegantly explains the ability of DTD to act on multiple D-amino acids, even when they are bound in a fixed orientation.

Based on their ‘atomic snapshots’, [3] had proposed a catalytic mechanism implicating the role of Thr90 in mounting a nucleophilic attack on the carbonyl carbon of the substrate. However, our structure clearly shows that the distance and orientation of the γ-hydroxyl group of Thr90 is unfavorable for the proposed nucleophilic attack (Figure 3—figure supplement 2). In order to experimentally rule out the earlier proposed catalytic mechanism, we have now mutated Thr90 to Ala in both *Pf*DTD and *Ec*DTD and show that the mutants are completely active (Figure 3—figure supplement 2).

With these additional analyses and data, we unequivocally rule out the earlier structural model both statistically and experimentally not only in terms of ligand binding mode but also in terms of the proposed catalytic mechanism. It is for the reasons above that the earlier studies could not identify the functionally critical residues and hence were unable to provide the structural basis for the chiral proofreading mechanism.

[Editors’ note: the author responses to the second round of peer review follow.]

*1) The authors now describe mutations that indicate that the binding mode they see is relevant to catalysis. They also carry out some mutations around the active site that support nucleotide-mediated catalysis, although this point in not rigorously proven because the data are negative (i.e., mutations do not affect catalysis substantially). Hence, they should be careful to tone down the discussion regarding previously solved structures of DTD with other ligands. The focus should be on what can be definitively concluded from the combination of structures presented here and other earlier structures rather than just over stating apparent ‘failures’ of either. This would allow for better discussion of the different catalytic mechanisms proposed from earlier structures obtained with other ligands and what is proposed to be the case here. Specifically, the Introduction should describe the earlier structural work and place the present work properly in context of the previous work, before moving on to a discussion of the present work*.

We agree with the reviewers that the discussions should be toned down on the apparent failure of the earlier work. Our original manuscript was along that line but to address certain criticisms during the review process we had incorporated those changes. However, we have now removed those parts as suggested. We have now modified the last paragraph in the Introduction section to describe all the previous structural work carried out on DTD and the information that was already available before the current study. We have also mentioned the proposals that were based on these structural studies in the same paragraph and then highlighted the findings of the current work, which the earlier reports did not provide, in the Discussion section. We have now condensed the first three paragraphs of the Discussion section to two in order to completely tone down the criticism of the earlier work. We have also removed the Table S4 from the supplementary material which was redundant in referring to the lacunae in the earlier work.

*2) It is very difficult to place the results discussed here in the context of DTD from various species (eukaryotes, prokaryotes, mammals...). The Abstract should state which species is analyzed, and the Introduction should also state more clearly the extent to which DTD is conserved in different species and which ones are analyzed here. It is true that this information is available in the manuscript, but it should be provided up front and in a way that the reader can immediately appreciate*.

In response to this concern, we have now incorporated the name of the species analyzed in the third sentence of the Abstract. We have also modified the Introduction section to comprehensively describe the distribution of DTD in the biological world. In the same paragraph, we have also added a statement on conservation of DTD sequence between *E. coli* and humans to indicate the extent of conservation of this enzyme. Moreover, we clearly state the species used for various analyses in the last paragraph of the Introduction section.

*3) The word “novel” should be deleted from the Abstract. It is not clear in what sense the solution that this enzyme has reached for specificity is “novel” and in any case the novelty should be left for others to judge*.

We have removed the word “novel” from the Abstract.

*4) In the Discussion of the active site mutants, it is stated “The above data clearly demonstrate that none of the protein residues around the scissile bond are involved in catalysis”. Since only negative data are presented, this statement is too strong. Change “clearly demonstrate” to “indicate” or some other appropriate word. Likewise, a little later the paper states “…the above data strongly suggest that the DTD fold is designed to be an RNA-based catalyst in the proofreading reaction”. Again, in the absence of positive data about the mechanism, replace “strongly suggest” by “are consistent with” or some other appropriate wording*.

We have changed “clearly demonstrate” to “suggest” as suggested by the reviewers. We have also changed “strongly suggest” to “indicate” as suggested.

*5) Towards the end of the paper it is said “These data, therefore, clearly show the lack of functional relevance of the binding modes proposed earlier...”. This is too strong. How do the authors know that the previously seen binding modes are not intermediates for some part of the process of catalysis? Remove this wording and replace with appropriate text*.

We have toned down the statement “These data, therefore, clearly show the lack of functional relevance of the binding modes proposed earlier...” by changing it to “These data, therefore, rule out the earlier propositions…”

*6) A little later, it is stated “More importantly, the current study identifies and reveals for the first time the key role of an invariant cross-subunit Gly-*cis*Pro motif in solving...”. Delete “and reveals for the first time”. This work is built on earlier structural work and it is sufficient to say that it identifies a key role for this motif*.

We have deleted “and reveals for the first time” as suggested by the reviewers.

*7) The paragraph in the Discussion beginning “The study opens up important questions...” should be deleted. It is too speculative and it is far from certain that the structure can be used to design DTD enzymes to create a “D-amino acid world”*.

We have deleted the concerned paragraph in the Discussion section and added a small part (which we felt was necessary to mention) in the last line of the Discussion.